# Autism and attention-deficit/hyperactivity disorder among individuals with a family history of alcohol use disorders

Jan Sundquist[1,2]*, Kristina Sundquist[1,2], Jianguang Ji[1]*

[1]Center for Primary Health Care Research, Lund University, Malmö, Sweden; [2]Stanford Prevention Research Center, Stanford University School of Medicine, Stanford, United States

**Abstract** Recent studies suggest de novo mutations may involve the pathogenesis of autism and attention-deficit/hyperactivity disorder (ADHD). Based on the evidence that excessive alcohol consumption may be associated with an increased rate of de novo mutations in germ cells (sperms or eggs), we examine here whether the risks of autism and ADHD are increased among individuals with a family history of alcohol use disorders (AUDs). The standardized incidence ratios (SIRs) of autism and ADHD among individuals with a biological parental history of AUDs were 1.39 (95% CI 1.34–1.44) and 2.19 (95% CI 2.15–2.23), respectively, compared to individuals without an affected parent. Among offspring whose parents were diagnosed with AUDs before their birth, the corresponding risks were 1.46 (95% CI 1.36–1.58) and 2.70 (95% CI 2.59–2.81), respectively. Our study calls for extra surveillance for children with a family history of AUDs, and further studies examining the underlying mechanisms are needed.

## Introduction

Autism is a neural development disorder characterized by impaired social interaction and verbal and non-verbal communication (*Biederman and Faraone, 2005*; *Levy et al., 2009*). Attention-deficit/hyperactivity disorder (ADHD) is characterized by inappropriate levels of inattention and hyperactivity, resulting in functional impairment (*Biederman and Faraone, 2005*). Autism and ADHD typically begin in childhood and often persist into adulthood (*Biederman and Faraone, 2005*; *Levy et al., 2009*). A recent study suggests that autism and ADHD share common genetic mutations (*Cross-Disorder Group of the Psychiatric Genomics Consortium, 2013*), which may provide the genetic basis for the associations of the two disorders. The etiology of autism and ADHD is, however, largely unknown, although a few genetic and environmental factors have been suggested to be associated with their developments (*Szpir, 2006*; *Walsh et al., 2008*). Epidemiological studies suggest that the prevalence of both disorders is increasing (*Fombonne, 2005*; *Parner et al., 2008*; *Boyle et al., 2011*), and that advanced parental age at the time of conception is associated with increased risk of autism (*Kong et al., 2012*; *Parner et al., 2012*). A subsequent study showed that the risk of autism was also higher in grandchildren with an older maternal grandmother compared to those with a younger one (*Golding et al., 2010*). In addition, twin studies showed that the concordance rate for autism is much higher in monozygotic twins than in dizygotic twins (*Nordenbaek et al., 2013*), suggesting de novo mutations may partly explain the development of autism (*Callaway, 2012*; *Kong et al., 2012*). Previous studies found that a number of de novo mutations and epigenetic modifications in germ cells could lead to increased rates of autism, schizophrenia, and other mental disorders (*Malaspina, 2001*; *Fombonne, 2005*; *Boyle et al., 2011*; *Callaway, 2012*; *Kong et al., 2012*; *Parner et al., 2012*; *Gratten et al., 2013*; *Zaidi et al., 2013*). It is thus important to identify potential environmental factors that contribute

***For correspondence:** jan.sundquist@med.lu.se (JS); jianguang.ji@med.lu.se (JJ)

**Competing interests:** The authors declare that no competing interests exist.

**eLife digest** Children learn to talk, manage their emotions, and control their behavior in a period when the brain is developing rapidly. The first signs of several developmental disorders, such as autism and attention-deficit/hyperactivity disorder (ADHD), may also emerge during this period. Children with autism may have difficulties with social interactions and communication, while those with attention-deficit/hyperactivity disorder may struggle to pay attention to a task and may be more active than other children.

Autism or ADHD are diagnosed based on the child's behavior because the underlying causes of the disorders are not well understood. Both genes and the environment have been linked to the conditions; and it was recently suggested that certain common genetic mutations are more common in children with ADHD or autism. However, as some of the mutations linked to autism are not found in the genes of the affected children's parents, it is likely that they occurred in either of the sperm or the egg cell from the parents.

Exposure to harmful substances in the environment can cause mutations in egg or sperm cells, or alter the expression of genes without changing the gene sequence. Excessive alcohol consumption is one environmental factor that can mutate genes or alter gene expression. Here, Sundquist et al. have looked to see if there is a relationship between a child having a parent with an alcohol use problem and the child's risk of developing autism or ADHD.

Examining national medical registries identified 24,157 people with autism and 49,348 with ADHD in Sweden between 1987 and 2010. Sundquist et al. discovered that autism and ADHD were more common in individuals who had a parent with a history of an alcohol use disorder than in those whose parents had no history of an alcohol use disorder. There was also an even greater risk of either condition if the parent had been diagnosed with an alcohol use problem before the birth of the child.

Adopted children who had a biological parent with an alcohol use disorder were at a greater risk of autism and ADHD than those whose adoptive parent had an alcohol use disorder. However, as very few adopted parents were diagnosed with an alcohol use problem, it is important to be cautious about drawing firm conclusions from this observation.

Sundquist et al. estimate that around 4% of autism cases and 11% of ADHD cases could be avoided if parents abstained from heavy alcohol consumption. Though these findings are consistent with parents with an alcohol use disorder being more likely to pass on mutations to their children, there are also other possible explanations. As such, further research examining the underlying cause is still needed.

to the increased rate of de novo mutations and/or epigenetic modifications in the parents (*Kinney et al., 2010*). We assume that excessive alcohol consumption may be associated with an increased rate of de novo mutations in germ cells (sperms or eggs), because previous studies showed that exposures to alcohol could lead to increased rates of mutations in germ cells (*Narod et al., 1988*; *Yamauchi et al., 2012*). It is thus necessary to examine whether the risks of autism and ADHD are increased among individuals with a family history of alcohol use disorders (AUDs). In addition, we examined the offspring risks of autism and ADHD when their adoptive parents were diagnosed with AUDs to disentangle the potential contribution of high levels of psychological tension and stress in AUD families.

## Results

In *Table 1*, we present the basic characteristics of individuals with autism and ADHD. A total of 24157 and 49348 individuals were identified with autism and ADHD in Sweden between 1987 and 2010. Men were more often diagnosed with these two disorders than women, and the median age at diagnosis was for both disorders 16 years. The incidences of the two disorders increased greatly during the last decade.

The risks of autism and ADHD among individuals with a family history of AUDs are presented in *Table 2*. During more than 10 million person-years of follow-up, a total of 3136 individuals were diagnosed with autism and 10,047 individuals were diagnosed with ADHD. The overall risks of autism and ADHD were 1.39 (95% CI 1.34–1.44) and 2.19 (95% CI 2.15–2.23), respectively, among individuals with

**Table 1.** Basic characteristics of patients with autism and ADHD

| Characteristic | | Autism | | ADHD | |
|---|---|---|---|---|---|
| | | No. | % | No. | % |
| Sex | Male | 16808 | 69.6 | 33491 | 67.9 |
| | Female | 7349 | 30.4 | 15857 | 32.1 |
| Age (years) | <10 | 5995 | 24.8 | 8129 | 16.5 |
| | 10–19 | 9671 | 40.0 | 22854 | 46.3 |
| | 20–29 | 4215 | 17.4 | 7727 | 15.7 |
| | 30+ | 4276 | 17.7 | 10638 | 21.6 |
| Time period | 1987–1990 | 425 | 1.8 | 207 | 0.4 |
| | 1991–1995 | 656 | 2.7 | 381 | 0.8 |
| | 1996–2000 | 1199 | 5.0 | 1307 | 2.6 |
| | 2001–2005 | 6457 | 26.7 | 9953 | 20.2 |
| | 2006–2010 | 15420 | 63.8 | 37500 | 76.0 |
| All | | 24157 | 100.0 | 49348 | 100.0 |

**Table 2.** Risk of autism and ADHD in offspring when their parents were diagnosed with alcohol use disorder (AUD)

| AUD in parent | No. of offspring at risk | Person-years of follow-up | Autism | | | | ADHD | | | |
|---|---|---|---|---|---|---|---|---|---|---|
| | | | O | SIR | 95% CI | | O | SIR | 95% CI | |
| Risk in sons | | | | | | | | | | |
| Father | 235696 | 4485707 | 1793 | **1.39** | 1.33 | 1.45 | 5548 | **2.17** | 2.11 | 2.22 |
| Mother | 69214 | 1344766 | 567 | **1.55** | 1.43 | 1.69 | 1952 | **2.73** | 2.61 | 2.85 |
| Parents | 289763 | 5540362 | 2203 | **1.41** | 1.35 | 1.47 | 6800 | **2.20** | 2.15 | 2.25 |
| Risk in daughters | | | | | | | | | | |
| Father | 225317 | 4306194 | 753 | **1.31** | 1.22 | 1.4 | 2609 | **2.12** | 2.04 | 2.21 |
| Mother | 65,479 | 1275507 | 245 | **1.50** | 1.32 | 1.7 | 970 | **2.77** | 2.60 | 2.95 |
| Parents | 276439 | 5305285 | 933 | **1.34** | 1.25 | 1.43 | 3247 | **2.18** | 2.10 | 2.25 |
| Risk in offspring | | | | | | | | | | |
| Father | 461013 | 8791901 | 2546 | **1.36** | 1.31 | 1.42 | 8157 | **2.15** | 2.11 | 2.20 |
| Mother | 134693 | 2620273 | 812 | **1.54** | 1.43 | 1.65 | 2922 | **2.74** | 2.64 | 2.84 |
| Parents | 566202 | 10845647 | 3136 | **1.39** | 1.34 | 1.44 | 10,047 | **2.19** | 2.15 | 2.23 |
| PAF | | | 3.6% | | | | 11% | | | |

O, observed number of cases; SIR, standardized incidence ratio; CI, confidence interval.

Bold type, 95% CI does not include 1.00.

PAF, population attributable fraction.

an affected parent with AUDs compared to those without an affected parent. The risks were similar in affected sons and daughters. The PAF was 3.6 and 11.0%, respectively, for autism and ADHD.

To examine whether the observed association in *Table 2* is possibly due to putative germ cell mutation in parents with alcohol consumption, we studied the risk of autism and ADHD among offspring whose parents were diagnosed with AUDs before their birth (*Table 3*). Only 95,003 offspring had a parent diagnosed with AUDs before their birth, which accounted for 16% of all offspring with a family history of AUDs. After 922,618 person-year of follow-up, the overall risks of autism and ADHD were 1.46 (95% CI 1.36–1.58) and 2.70 (95% CI 2.59–2.81), respectively, as compared to those without an affected parent. The increase risks were similar in affected sons and daughters.

In *Table 4*, we present the risk of autism and ADHD among adoptees when either their biological or adoptive parents were diagnosed with AUDs. The risks of autism and ADHD were significantly

**Table 3.** Risk of autism and ADHD in offspring when their parents were diagnosed with alcohol use disorder (AUD) before the birth of the offspring

| AUD in parent | No. of offspring at risk | Person-years of follow-up | Autism | | | ADHD | | |
|---|---|---|---|---|---|---|---|---|
| | | | O | SIR | 95% CI | O | SIR | 95% CI |
| Risk in sons | | | | | | | | |
| Father | 41832 | 410327 | 442 | **1.44** | 1.31 1.59 | 1555 | **2.60** | 2.48 2.74 |
| Mother | 8776 | 80,689 | 99 | **1.67** | 1.35 2.03 | 347 | **3.14** | 2.82 3.49 |
| Parents | 48704 | 472596 | 514 | **1.46** | 1.34 1.59 | 1778 | **2.61** | 2.49 2.73 |
| Risk in daughters | | | | | | | | |
| Father | 39684 | 390429 | 145 | **1.40** | 1.18 1.64 | 565 | **3.05** | 2.80 3.31 |
| Mother | 8457 | 77259 | 36 | **1.79** | 1.25 2.48 | 119 | **3.44** | 2.85 4.12 |
| Parents | 46299 | 450022 | 175 | **1.47** | 1.26 1.70 | 634 | **3.00** | 2.77 3.24 |
| Risk in offspring | | | | | | | | |
| Father | 81516 | 800756 | 587 | **1.43** | 1.32 1.55 | 2120 | **2.71** | 2.59 2.83 |
| Mother | 17233 | 157948 | 135 | **1.70** | 1.42 2.01 | 466 | **3.22** | 2.93 3.52 |
| Parents | 95003 | 922618 | 689 | **1.46** | 1.36 1.58 | 2412 | **2.70** | 2.59 2.81 |

O, observed number of cases; SIR, standardized incidence ratio; CI, confidence interval.
Bold type, 95% CI does not include 1.00.

**Table 4.** Risk of autism and ADHD in adoptees when their biological or adoptive parents were diagnosed with alcohol use disorder (AUD)

| | No. of offspring at risk | Person-years of follow-up | Autism | | | ADHD | | |
|---|---|---|---|---|---|---|---|---|
| | | | O | SIR | 95% CI | O | SIR | 95% CI |
| Biological parents with AUD | | | | | | | | |
| Father | 2175 | 51746 | 16 | **2.15** | 1.23 3.50 | 53 | **2.63** | 1.97 3.44 |
| Mother | 1381 | 32745 | 10 | 1.81 | 0.86 3.35 | 18 | 1.21 | 0.72 1.92 |
| Parents | 3251 | 77310 | 19 | **1.75** | 1.05 2.74 | 58 | **1.91** | 1.45 2.47 |
| Adoptive parents with AUD | | | | | | | | |
| Father | 457 | 10,839 | 1 | 0.75 | 0.00 4.28 | 7 | 1.78 | 0.70 3.68 |
| Mother | 229 | 5443 | 0 | | | 4 | 1.93 | 0.50 4.98 |
| Parents | 654 | 15546 | 1 | 0.51 | 0.00 2.93 | 9 | 1.59 | 0.72 3.04 |

O, observed number of cases; SIR, standardized incidence ratio; CI, confidence interval.
Bold type, 95% CI does not include 1.00.

increased when their biological parents were diagnosed with AUDs with a SIR of 1.75 and 1.91, respectively. However, the risks were not significant when their adoptive parents were diagnosed with AUDs, possibly because of limited numbers of cases.

## Discussion

In this population-based cohort study from Sweden of 24,157 and 49,348 individuals with autism and ADHD, respectively, we found that the risks were significantly increased among individuals with a family history of AUDs, after adjusting for potential confounding factors. The incidence rates of autism and ADHD were somewhat more pronounced when the diagnosis of AUDs in parents occurred before the birth of their offspring. The time-relationship supports that the observed association between parental AUDs and autism and ADHD in offspring may be causal. In addition, we studied the incidence among adoptees to disentangle the contribution by environmental factors. However, we could not draw definite conclusions from these results, as the cases were very few among adoptees with an affected adoptive parent. The PAF was 3.6 and 11.0% for autism and ADHD, respectively, if excess alcohol consumption could be avoided in their parents.

There are a few possible biological explanations for the high observed risks of autism and ADHD among individuals with a family history of AUDs. Firstly, they could be explained by an increased rate of de novo mutations among individuals with AUDs, confirming our hypothesis. It should be noted that although a few de novo mutations have been found to be related to autism (*Callaway, 2012*; *Kong et al., 2012*; *Zaidi et al., 2013*), associations with ADHD are still lacking, calling for more research to examine possible mutations among patients with ADHD. However, a recent study found that autism and ADHD share common genetic mutations (*Cross-Disorder Group of the Psychiatric Genomics Consortium, 2013*), which may partly explain their associations with parental AUD. In addition, the increased risks of autism and ADHD may be related to epigenetic modifications of specific genes because exposure to alcohol can alter the methylation status in sperm cells (*Govorko et al., 2012*). How epigenetic modifications may affect the development of these disorders should, however, be examined in further studies. It should be noted that there might be common genetic predispositions linked to AUD, autism and ADHD. An additional explanation for the observed associations is that children of individuals with alcohol problems may face high levels of tension and stress at home, and that a stressful home environment may lead to communication problems, bad school performances, and, ultimately, to the development of autism and ADHD (*Berger, 1993*). However, our study found that the risk of autism and ADHD was increased only when the biological parents were diagnosed with AUDs, suggesting that genes as well as a stressful home environment have an impact on the increased risk of autism and ADHD.

One possible reason for the high risks of autism and ADHD among offspring with mothers with AUDs is that mothers with AUDs may continue drinking alcohol during pregnancy, causing long-term structural, behavioral, and cognitive damage to their children. It is known that alcohol can be carried to the placenta and that the concentration of alcohol in the unborn baby's bloodstream is the same as that in the mother. Mothers who consume alcohol during pregnancy may give birth to a baby with fetal alcohol syndrome (FAS). In general, the more severe the mother's drinking problem is during pregnancy, the more severe are the symptoms of FAS in infants (*Landgren et al., 2010*). Children with FAS may be diagnosed with autism or ADHD because these conditions are all characterized by an impaired quality of social interaction (*Bishop et al., 2007*).

Around 4% of autism and 11% of ADHD in offspring can be avoided if parents avoid excessive alcohol consumption. It should be noted that even moderate alcohol consumption during pregnancy may affect the child's cognitive score (*Lewis et al., 2012*), suggesting that the proportion of the two diseases related to parental alcohol consumption would be even higher if individuals with moderate alcohol drinking were also included in the calculation of PAF.

An important strength of this study is that all the data were retrieved from Swedish registers with high quality and coverage. In addition, the number of patients included is large enough to guarantee reliable risk estimates. The prospective study design and the completeness of the follow-up of patients are other major advantages of the present study. Moreover, we adjusted for a number of confounding factors and thereby minimized confounding.

One limitation of this study is that only individuals with autism and ADHD who visited either primary health care or a hospital were included. Another limitation is the lack of information on some individual-level risk factors, such as parental drinking habits of moderate nature, psychosocial factors, and sociocultural behaviors. In addition, it is likely that the identification of AUD in parents could be underreported.

In summary, individuals with a family history of AUDs had increased risks of autism and ADHD, which calls for further study to examine the underlying mechanisms and for clinical attention and extra surveillance for children with a family history of AUDs.

## Materials and methods

This study was approved by the Ethics Committee at Lund University, Sweden. Individuals with autism and ADHD were identified from four Swedish Registers: the primary health care registers covering the counties of Skåne (1987–2010) and Stockholm (2001–2007) and the National Swedish Hospital Discharge Register (1987–2010) and Outpatient Register (2001–2010). The primary health care register in Skåne (the southernmost province of Sweden), PASIS, contains data on individuals (around 1.2 million) who visited primary health care in Skåne between 1987 and 2010. The primary health care register in Stockholm covers 75 primary health care centers for the period 2001–2007. The Swedish Hospital Discharge Register was founded in 1964–65 by the National Board of Health and Welfare, and

has had complete nationwide coverage since 1987. The Outpatient Register contains data on all visits to outpatient clinics in Sweden since 2001. A recent external review suggested that the overall accuracy of the Swedish Hospital Registry is approximately 85–95% (*Ludvigsson et al., 2011*). Individuals with autism were identified according to the International Classification of Diseases (ICD-9 code 299B and ICD-10 code F840). Individuals with ADHD were identified according to ICD-9 code 314 and ICD-10 code F90.

We further linked individuals with autism and ADHD to the Swedish Multi-Generation Register to identify their parents (*Ekbom, 2011*). The Swedish Multi-Generation Register contains data on the biological parents of index persons born in or since 1932 and registered in Sweden since 1961. The Multi-Generation Register contains data on more than 3.2 million families and 12 million individuals. In addition, we identified 22,996 individuals who were born after 1950 and had been adopted, with information available on both adoptive parents and at least one biological parent. Adoptees adopted by biological relatives or by an adoptive parent living with a biological parent were excluded. Diagnoses of AUDs in parents (biological or adoptive) were identified from three national registries, including the Hospital Discharge and Outpatient Registers using the ICD-9 (291A–291F, 291W, 291X, 303, 305A) and ICD-10 codes (F10), the Crime Register (which contains data on individuals who committed AUD-related crimes in 1973–2007), and the Swedish Pharmacy Register (2005–2010). Data from the latter were based on prescriptions of disulfiram, naltrexone, and acamprosate. Using these three sources, we identified 420,489 unique individuals with AUDs (lifetime prevalence in Sweden of ~3.8%).

Additional linkages were made to the Statistics Sweden's Total Population Register (≥1990) and the Population a Housing Census (<1990) to obtain information on individual-level characteristics, such as year of birth, sex, and country of birth; to the Cause of Death Register to identify date of death; and to the Emigration Registry to identify date of emigration. All linkages were performed using individual national identification numbers, which were replaced with serial numbers in order to preserve anonymity.

We calculated person-years at risk of autism and ADHD among individuals with a family history of AUDs (either biological or adoptive parents with AUDs) from the date of birth, immigration, or 1 January 1987, whichever came last, until the date of diagnosis of autism or ADHD, death, emigration, or the end of the study period (31 December 2010), whichever came first. Standardized incidence ratios (SIRs) were calculated as the ratio of the observed and expected numbers of cases (*Breslow and Day, 1987*; *Rothman and Greenland, 1998*). The expected number of cases was calculated according to the incidence rate for all individuals without a family history of AUDs. SIRs were standardized by 5-year age group, sex (male and female), 5-year time period, family history of autism or ADHD, and individual disposable income (*Esteve et al., 1994*). 95% confidence intervals for the SIRs were calculated assuming a Poisson distribution, and were rounded to the nearest two decimals (*Esteve et al., 1994*). The population attributable fraction (PAF) is the proportion of the disease burden in the population that could be prevented if exposure to a particular risk factor could be avoided (*dos Santos Silva, 1999*). The calculation of PAF is based on a relative risk estimate. In this study, SIRs were used to calculate PAF as follows: PAF = proportion of autism and ADHD patients with a family history of AUD × (SIR − 1)/SIR. All analyses were performed using SAS version 9.1 (SAS Institute, Cary, NC).

## Acknowledgements

The authors wish to thank the CPF's Science Editor Stephen Gilliver for his valuable comments on the text.

---

## Additional information

### Funding

| Funder | Grant reference number | Author |
| --- | --- | --- |
| Vetenskapsrådet | 2011-3340 | Kristina Sundquist |
| Forte: Swedish Research Council for Health, Working Life and Welfare | 2013-1836 | Kristina Sundquist |
| Region Skåne | ALF funding | Jianguang Ji |
| Fredrik och Ingrid Thurings Stiftelse | ALF funding | Jianguang Ji |

| Funder | Grant reference number | Author |
|---|---|---|
| Vetenskapsrådet | 2012-2378 | Jan Sundquist |
| National Institute of Drug Abuse | R01 DA030005 | Jan Sundquist |

The funders had no role in study design, data collection and interpretation, or the decision to submit the work for publication.

## Author contributions

JS, KS, Acquisition of data, Drafting or revising the article, Contributed unpublished essential data or reagents; JJ, Conception and design, Acquisition of data, Analysis and interpretation of data, Drafting or revising the article, Contributed unpublished essential data or reagents

## Ethics

Human subjects: This study was approved by the Regional Ethical Review Board of Lund University (reference number 2008-409) in Sweden.

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
