## [Decision Letter]

Thank you for sending your work entitled “Autism and Attention-Deficit/Hyperactivity Disorder among Individuals with a Family History of Alcohol Use Disorders” for consideration at *eLife*. Your article has been favorably evaluated by Prabhat Jha (Senior editor), a Reviewing editor, and 2 reviewers.

The Reviewing editor and the reviewers discussed their comments before we reached this decision, and the Reviewing editor has assembled the following comments to help you prepare a revised submission.

Your paper presents intriguing data on the association between a diagnosis of alcoholism in at least one of the parents and autism or ADHD in the offspring. The following points must be addressed in your revision. Although all six items are important we consider concern #1 below as absolutely critical. You must convince us that the preponderance of the evidence is in line with your hypothesis, i.e., that the putative germ line mutations induced by alcohol occurred before the birth of the child. Our decision to publish will primarily hinge on this demonstration.

1) The most important concern we have is that you may have misinterpreted the directionality of the causal relation. How do you know that the putative germ cell mutations that you attribute to heavy and chronic alcohol consumption in the parents occurred before the conception of the affected offspring? You must carry out analyses and add evidence to demonstrate that the diagnosis of alcoholism (or at least the onset of the excessive consumption that led to a clinical diagnosis eventually) occurred before the birth of the child. This is the only analytical framework that can bolster your underlying hypothesis. Having a child affected by autism or ADHD can lead parents to seek emotional relief in alcohol drinking, which would also create a statistical association. Therefore, it is paramount that you document the temporality of the association.

2) Although the attempt to distinguish between risks in biological and adoptive parents is commendable much of your reasoning hinges on the lack of significance among the latter to advance the notion of a causal mechanism. With at most nine parents for ADHD and one father for autism the analysis for cases among adoptive parents is imprecise, and thus meaningless. You must refrain from making sweeping statements about the differences between findings in biological and adoptive parents as being in support of your hypothesis.

3) The argument that the effect is greater for maternal than paternal alcoholism calls for a specific test for effect modification for the data summarized in Table 2.

4) Tables 2, 3 and 4 should be more explicit and indicate in additional columns the number of persons at risk and hopefully the person-time accumulated in each category.

5) Table 3 needs to be better explained. It seems that your study did not confirm the expected trend of increased autism risk with advanced paternal age.

---

## [Author Response]

*1) The most important concern we have is that you may have misinterpreted the directionality of the causal relation. How do you know that the putative germ cell mutations that you attribute to heavy and chronic alcohol consumption in the parents occurred before the conception of the affected offspring? You must carry out analyses and add evidence to demonstrate that the diagnosis of alcoholism (or at least the onset of the excessive consumption that led to a clinical diagnosis eventually) occurred before the birth of the child. This is the only analytical framework that can bolster your underlying hypothesis. Having a child affected by autism or ADHD can lead parents to seek emotional relief in alcohol drinking, which would also create a statistical association. Therefore, it is paramount that you document the temporality of the association*.

This is a good point. We have now also calculated the risk of autism and ADHD in offspring when their parents were diagnosed with alcohol use disorder (AUDs) before the birth of the offspring, and the results are shown in Table 3.

*2) Although the attempt to distinguish between risks in biological and adoptive parents is commendable much of your reasoning hinges on the lack of significance among the latter to advance the notion of a causal mechanism. With at most nine parents for ADHD and one father for autism the analysis for cases among adoptive parents is imprecise, and thus meaningless. You must refrain from making sweeping statements about the differences between findings in biological and adoptive parents as being in support of your hypothesis*.

Thanks for this comment. We have modified the Discussion section to avoid any imprecise conclusions.

*3) The argument that the effect is greater for maternal than paternal alcoholism calls for a specific test for effect modification for the data summarized in*
Table 2.

We agree with this comment. However, the differences in offspring risk between maternal and paternal alcoholism are rather small and probably not clinically meaningful. We have therefore omitted the argument that the effect is greater for maternal as opposed to paternal alcoholism.

*4)*
Tables 2, 3 and 4
*should be more explicit and indicate in additional columns the number of persons at risk and hopefully the person-time accumulated in each category*.

This is a good point. We have included the number of individuals at risk and the person–years of follow-up in these tables.

*5)*
Table 3
*needs to be better explained. It seems that your study did not confirm the expected trend of increased autism risk with advanced paternal age*.

This is correct. We could not confirm an increased risk of autism with advanced paternal age, and we have deleted this table because our study focused on the risk of autisms and ADHD among offspring with a family history of AUDs rather than the potential effect of advanced parental age on autism risk.